Factors influencing space-use and kill distribution of sympatric lion prides in a semi-arid savanna landscape

http://orcid.org/0000-0003-4623-6097 Tarugara Allan allan@malilangwe.org
Clegg Bruce W.
Clegg Sarah B.
Research Department, Malilangwe Wildlife Reserve , Chiredzi, Masvingo , Zimbabwe
Hedrick Ann
Electronic publication date: 2024 Jan 24
Publication date: 2024
Volume: 12
Electronic Location ID: e16749
Received 2023 Sep 7; Accepted 2023 Dec 13
Copyright: © 2024 Tarugara et al.
Copyright year: 2024
Copyright holder: Tarugara et al.
License: This is an open access article distributed under the terms of the Creative Commons Attribution License, which permits unrestricted use, distribution, reproduction and adaptation in any medium and for any purpose provided that it is properly attributed. For attribution, the original author(s), title, publication source (PeerJ) and either DOI or URL of the article must be cited.
License URL: https://creativecommons.org/licenses/by/4.0/

Keywords: Competition, Habitat selection, Hierarchy, Home range, Panthera leo, Surface water

Funding: The Malilangwe Trust, Zimbabwe Funding for this work was provided by the Malilangwe Trust, Zimbabwe. The funders had no role in study design, data collection and analysis, decision to publish, or preparation of the manuscript.

==============================
Understanding lions’ (Panthera leo) space-use is important for the management of multi-species wildlife systems because lions can have profound impacts on ecosystem-wide ecological processes. Semi-arid savanna landscapes are typically heterogeneous with species space-use driven by the availability and distribution of resources. Previous studies have demonstrated that lions select areas close to water as encounter rates with prey are higher and hunting success is greater in these regions. Where multiple lion prides exist, landscape partitioning is expected to follow a despotic distribution in which competitively superior prides occupy high-quality areas while subordinates select poorer habitats. In this study, Global Positioning System collar data and logistic regression were used to investigate space-use and hunting success among 50% of lion prides at Malilangwe Wildlife Reserve, Zimbabwe. Our findings show that lion space-use was driven by surface water availability and that home range selection was socially hierarchical with the dominant pride occupying habitat in which water was most abundant. In addition, we found that the effect of shrub cover, clay content and soil depth on kill probability was area specific and not influenced by hierarchical dominance. Where multiple lion prides are studied, we recommend treating prides as individual units because pooling data may obscure site and pride specific response patterns.

Introduction

Understanding why animals occur where they do is a cornerstone of ecology (Burt, 1943; Krebs, 1980; Powers & McKee, 1994). Where animals spend most of their time in a habitat is determined by a complex interplay of social and environmental factors (Davies et al., 2016; Moyer, McCown & Oli, 2008). Prey availability and catchability are major factors driving patterns of space-use among large carnivores (Davidson et al., 2012; Hopcraft, Sinclair & Packer, 2005; Ogutu & Dublin, 2004), but for social species, access to profitable areas may be moderated by social pressures (Struhsaker, 1967). Lions (Panthera leo) are an important focal species in most conservation and tourism-based systems (Loveridge et al., 2009) and as an apex predator, they have profound impacts on ecosystem-wide ecological processes and, therefore, understanding their space-use patterns and predation dynamics is important for informing management decisions on carrying capacities and herbivore stocking rates (Webb, Hebblewhite & Merrill, 2008; Tambling et al., 2010; McPhee, Webb & Merrill, 2012). Lion spatial and foraging ecology in savanna ecosystems has been extensively researched yet new studies are warranted to broaden the body of knowledge. This is because, generalizations can be misleading because prey assemblages, social and demographic factors as well as environmental conditions differ between prides, home ranges and protected areas. As such, studies that consider both environmental and social factors at the inter-pride level could provide valuable insights into lion behaviour.

Semi-arid savanna landscapes are typically heterogeneous, with resources patchily distributed and habitats varying in quality (Gaillard, Festa-Bianchet & Yoccoz, 1998; Jetz et al., 2004). Habitat quality is determined by environmental conditions with surface water availability being the primary factor limiting the distribution of herbivore populations (Owen-Smith, 1996; Redfern et al., 2003; Valeix et al., 2009). Most herbivores need to drink regularly and, as such, areas close to water sources are usually associated with large aggregations of prey and, in turn, attract predators (de Boer et al., 2010; Cain, Owen-Smith & Macandza, 2012). Several studies have demonstrated that lions select habitats close to water as these provide increased encounters with prey and associated hunting success (Duncan et al., 2012; Harrington et al., 1999; Kittle et al., 2022; Owen-Smith, 1996; Valeix et al., 2010). Such areas are profitable to lions as they provide the least energetic cost to acquire food (Mitchell & Powell, 2004). Where multiple prides exist, landscape partitioning is expected to follow an ideal despotic distribution (Fretwell & Lucas, 1969) in which competitively superior prides occupy the most profitable areas forcing subordinates into sub-optimal environments. When this occurs, prides are expected to respond to environmental determinants of space-use in accordance with their position in the social hierarchy. We hypothesize that distance from permanent surface water has a strong negative effect for the dominant pride, a medium negative effect for prides mid-way in the social hierarchy, and a weak negative or even a positive effect for the most subordinate prides, meaning that dominant prides are expected to be positively associated with surface water.

Among sympatric lion prides, social hierarchy is shaped by agonism (Heinsohn, 1997). The outcome of agonistic encounters depends on the strength of contending prides (Packer, Scheel & Pusey, 1990; Heinsohn, 1997). Previous studies have shown that lion group size plays an important role in shaping hierarchy dynamics, with large prides routinely outcompeting smaller ones (Hamilton, 1971; Alexander, 1974; Wilson, 1975; Packer, Scheel & Pusey, 1990). In addition, pride strength is determined by the age-sex composition of the pride (Mosser & Packer, 2009). For example, prides containing several adult males or females have a competitive advantage over those largely composed of younger, less experienced and smaller individuals (Packer, Scheel & Pusey, 1990; Mosser & Packer, 2009; Chakrabarti et al., 2020).

This study posits that broad-scale (study area) home range choice and fine-scale (kill site) hunting success of lions at Malilangwe Wildlife Reserve (MWR), Zimbabwe are influenced by both environmental and social factors. Being ambush predators, lions use the cover of vegetation when stalking prey (Hopcraft, Sinclair & Packer, 2005; Davidson et al., 2012) with areas containing sufficient cover improving hunting success. Soil depth and clay content determine plant available moisture and nutrients which, in turn, influences forage quality (Walker & Langridge, 1997). Deeper soils can contain large amounts of water and retain it for longer (Calviño, Andrade & Sadras, 2003; Tromp-van Meerveld & McDonnell, 2006; Clegg & O’Connor, 2017) while soils with a high clay content release more nutrients ensuring healthy plant growth (Tahir & Marschner, 2016; Clegg & O’Connor, 2017; Kome et al., 2019). As such, habitats with deeper soils and high clay content can provide high quality forage for extended periods thus attracting herbivore populations and, consequently, occupation by lions. The specific objectives of this study were to examine the functional relationships between lion home range selection and hunting success, and likely determinants of prey density viz. (i) distance to permanent surface water, (ii) shrub canopy cover and (iii) soil properties.

Method

Study area

The study was conducted at MWR which is located in the south-eastern lowveld of Zimbabwe between 20°58′ and 21°15′S and 31°47′ and 32°01′E (Fig. 1). MWR is a non-hunting property whose main objectives are conservation and community development (Ball et al., 2019). The reserve is approximately 490 km2 in size and altitude ranges from 290 m, in river systems, to 500 m above sea level on hills (Traill & Bigalke, 2007). Rainfall (mean ≈ 560 mm per year) is seasonal with approximately 84% of precipitation occurring between November and March and the average minimum and maximum monthly temperatures ranging from 13.4 °C (July) to 23.7 °C (December) and 23.2 °C (June) to 33.9 °C (November), respectively (Clegg & O’Connor, 2012). The southern boundary, bordering Gonarezhou National Park has a short fence that allows some movement of carnivores and smaller ungulates but restricts the movement of large herbivores.

Figure 1 Location of the study area.

Map created with QGIS v3.26. Map hillshade credit: QGIS v3.26.

Although soils vary, they are principally derived from alluvium, sandstone, gneiss and basalt (Clegg & O’Connor, 2012). Vegetation types at MWR can be broadly classified as riverine, hill miombo, mopane (Colophospermum mopane) veld, thorn thicket and open woodland (Clegg & O’Connor, 2012). Hills are largely dominated by redwood (Brachystegia tamarindiodes) and mnondo (Julbernardia globiflora) tree species, while low lying areas are characterized by mixed broadleaf woodland. The grass layer on MWR is mostly composed of Urochloa mossambicensis, Heteropogon contortus, Digitaria eriantha and Aristida spp. A wide variety of prey species inhabit the heterogenous landscape including Cape buffalo (Syncerus caffer), zebra (Equus burchellii) and impala (Aepyceros melampus) among other antelope species. Competing predators of lions include leopards (Panthera pardus), spotted hyenas (Crocuta crocuta) and wild dogs (Lycaon pictus).

Collaring of animals

Three independent lion prides (Banyini n = 12 (adult males = 2, adult females = 4, subadults = 6), Nduna n = 6 (adult males = 1, adult females = 2, subadults = 3) and Hlamba mlonga n = 4 (adult females = 2, subadults = 2)) were identified for collaring at MWR. Between February 21 and August 02, 2004, three GPS/drop-off collars (Televilt, Lindesberg, Sweden) were fitted onto either an adult male or female member of each pride. Lions were chemically immobilized with a combination of Zoletil-Medetomidine (1.0–20 mg/kg body mass), with the anaesthetic being darted into the muscular region of the hindquarters. Reversal was achieved by subcutaneous injection of Antisedan (at 2.5 mg/mg of Medetomidine) or Yohimbine (at 1 ml/50 kg of body weight) and lions were monitored until they fully recovered from the effects of the anaesthesia. Safe, professional, and humane animal care guidelines stipulated by the Scientific Experiments on Animals Act of Zimbabwe (Chapter 19:12 of 1963) and Olfert, Cross & McWilliam (1993) were adhered to. The National Animal Research Ethics Committee of Zimbabwe approved the study protocols (permit number: NAREC/008/23) and all animal handling procedures were performed by a licensed practitioner (with Zimbabwean Dangerous Drugs License number: 034/2004). Three other lion prides were present in the study area (Matsanga (n = 10, north-western section), Chiloveka (n = 8, south-western section) and Tennis Court (n = 6, south-eastern section)) over the period under investigation but were not included in this study.

Collar and kill data collection

Collars were programmed to fix a GPS position at 30-min intervals from 16:15 to 07:15 and at two 3-h intervals during the hottest part of the day, when lions are presumed to be less active (Kingdon et al., 2013) (i.e., 33 positions per day). A potential kill site was defined as a cluster of ≥3 consecutive GPS positions located within a 50 m radius of each other i.e., a location where a pride spent at least one and a half hours without moving (Tambling et al., 2010). A single point that was central within the cluster was chosen as a reference for the location of the site. The coordinates of potential kill sites were relayed to field scouts who visited the locations. At the site, the surroundings (50 m radius) were inspected for evidence of a kill. Data recorded included the presence or absence of a kill, the species of animal killed, identification method (bones, fur, horns etc.). Time since kill and possible pride composition were estimated from expert opinion by experienced field technicians.

Data analysis

This study postulates that space-use and hunting success of lions at MWR are influenced by both environmental and social factors. Lion space-use generally matches the distribution of their prey (Heithaus, 2001; Vanak et al., 2013) which, in turn, is shaped by surface water availability, vegetation cover and range condition (previous studies by Clegg & O’Connor (2012, 2017) showed that soil depth and clay content were the main gradients explaining the composition and structure of vegetation at MWR). Species-wide studies have demonstrated that large groups generally outcompete smaller ones (Adams, 2001; Carlson, 1986; Cheney, 1992; Grinnell, Packer & Pusey, 1995; Packer, Scheel & Pusey, 1990; Wilson & Wrangham, 2003) and occupy higher quality territories (Woolfenden & Fitzpatrick, 1984; Kauffman et al., 2007; Mosser & Packer, 2009) therefore we used pride size as a proxy for social dominance. We predicted that the Banyini pride would be dominant as it was twice the size of the next largest pride and had the highest number of adult males and females.

Landscape level home range selection

Minimum convex polygons (MCPs) delineating each lion pride’s home range were constructed using GPS presence data in Quantum GIS v3.26 (QGIS Development Team, 2022) and their sizes calculated. Logistic regression was used for analysis but because input data needs to be binomial (McCullagh & Nelder, 1989) and GPS collars only provide presence information, “pseudo-absence” points were generated to produce a presence-absence dataset (Odendaal-Holmes, Marshal & Parrini, 2014). To determine which environmental parameters were important for home range selection, each pride’s location fixes were overlaid onto MWR’s base map as training sites for spatial distribution modelling in MaxEnt v3.4 (Phillips, Anderson & Schapire, 2006). MaxEnt uses presence data together with environmental variables to model an approximation of a species’ niche within the geographical bounds of its environment (Phillips, Anderson & Schapire, 2006). Presence locations and the associated environmental attributes (distance to surface water, shrub canopy volume, soil depth, clay content) were used to model the suitability of MWR’s landscape in relation to each pride’s space-use choices. Areas classified as suitable were those in which 95% of a pride’s presence positions were recorded. Using this threshold, an environmental suitability map was created for each pride and reclassed to produce two categories: suitable and unsuitable.

Using QGIS, a set of spatially independent randomly distributed points (absence data) equal to the number of each pride’s presence points was generated within the unsuitable habitat of each pride. The full extent of MWR’s landscape was considered available to each pride when generating the absence data. Next, a raster surface of permanent water locations across the study area was created by mapping point (permanent pans, springs and dams) and linear (perennial streams and rivers) sources from digital aerial photographs and the DISTANCE module of TerrSet (Eastman, 2015) was used to calculate Euclidean distances between the presence and absence points and their nearest water sources. Similarly, data for shrub canopy volume, soil depth and clay content were extracted for the presence and absence points using respective maps created from a 2003 dataset (see Clegg & O’Connor (2012) for how these maps were created).

The lme4 package of the R statistical platform was used to perform logistic regression analysis to determine the effects of the environmental variables on the outcome of each pride’s ranging behaviour (Bates et al., 2014; R Development Core Team, 2017). Presence probability (presence or absence) was used as a response variable while distance to water, shrub canopy volume, soil depth and clay content were treated as fixed effects:

Presence probability ~ distance to water + shrub canopy volume + soil depth + clay content.

From model results, significance values (at α = 0.05) were used to identify environmental factors that were important in determining the ranging choices of each pride and the relationships were plotted using marginal effects. Environmental variables that were considered important were those that were statistically significant and elicited consistent responses across all prides.

Home range level kill site selection

Within each pride’s home range, we theorized that the location of sites where lions made kills was not random but that kill locations were determined by a combination of favourable environmental conditions. Following the methodology described above, kill data and environmental variables were used in MaxEnt to model fine-scale environmental suitability for kill success for each lion pride. An equal number of pseudo-absence points as kill locations were placed in the zone which lacked kills and values of distance to water, shrub canopy volume, soil depth and clay content extracted for use in logistic regression with kill probability being used as the response variable.

Prey encounter within home ranges

The rate at which lions encounter prey within an environment has an influence on hunting success (Cosner et al., 1999; Hayward et al., 2011). We tested the assumption that prey encounter rates (number of groups identified per search effort) were positively correlated to lion kill rates, with home ranges having high encounter rates expected to have correspondingly higher kill rates (Scheel, 1993; Nachman, 2006; Fryxell et al., 2007). Prey species population and distribution data were obtained from MWR’s 2004 census data. Malilangwe Wildlife Reserve conducts annual aerial surveys where the entire property is flown by helicopter along predetermined transects and animal numbers are counted (Clegg, 2011). The number of groups per unit area of prey species within lions’ preferred weight range (Hayward & Kerley, 2005) encountered during the survey were recorded and prey encounter rates were derived for each pride’s home range for comparison with lion hunting success and space-use.

Results

A total of 2,994 presence points (Banyini n = 1,935, Nduna n = 323, Hlamba mlonga n = 736) were recorded over the study period. Out of these, 253 clusters (Banyini n = 116, Hlamba mlonga n = 78, Nduna n = 59) were identified and visited, and evidence of kills was found at 146 sites (Banyini n = 67, Nduna n = 42, Hlamba mlonga n = 37), representing a 58% discovery rate (Fig. 2). The Banyini pride, being the largest in size and having the most adults, was considered most dominant, followed by the Nduna pride and lastly, the Hlamba mlonga pride which was smallest in size and had the least adults and no males. Analysis showed that the Banyini pride occupied a home range in which the average distance to water was shortest, while the Hlamba mlonga pride was found where the average distance to water was longest (Table 1). This outcome was expected. Intervals between kills decreased with increasing pride size and home range size was negatively correlated with pride size (Fig. 3). This indicates that where prey encounter rates, and by extension, prey densities are high (e.g., near surface water), kill rates are also high and small home ranges are capable of supporting large prides.

Figure 2 Map showing the location of home ranges and kill locations of the three study lion prides.

Map created with QGIS v3.26.

Table 1 Home range characteristics for the three study lion prides.

Pride	Pride size (n)	Home range size (ha)	Average distance to water (km)	Average shrub canopy cover (m3/ha)	Average clay content (%)	Average soil depth (m)	Game encounter rate (groups/km2)	Interval between kills (days)	
Banyini	12	13,035	1.1	8,537	21	77	0.99	6.3	
Nduna	6	15,259	1.3	7,843	14	66	0.87	7.7	
Hlamba mlonga	4	17,460	2.4	5,876	20	71	0.81	11.1	

Figure 3 Contour plots showing the relationships between pride size, home range size, interval between kills and prey encounter rate.

Influence of environmental factors on home range selection

The Banyini pride showed a strong (P < 0.01) negative response to increasing distance from surface water (most presence points for this pride were <4 km from water) (Table 2, Fig. 4). The Nduna pride, which was next in size, also showed a negative response to increasing distance from water but the effect was weaker (P < 0.15). In contrast, the Hlamba mlonga pride, which was the smallest in size, showed a strong (P < 0.01) positive response to distance from water. The change in direction and strength of the response across the prides was consistent with our predictions of how pride size would modify the response to an environmental variable, with the largest pride selecting habitats closest to surface water and the smallest pride relegated to marginal landscapes where surface water was relatively scarce, and the middle pride occupying a home range in-between.

Table 2 Results of logistic regression analysis.

	Fixed effects	Estimate	Std. Error	z value	Pr(>|z|)	
Banyini pride	Intercept	0.05	0.15	0.36	0.720	
distance to water	−9e−04	4e−05	−21.67	2e−16	
shrub canopy cover	−4e−05	1e−05	−4.26	2e−05	
clay content	0.01	4e−03	1.356	0.175	
soil depth	0.02	9e−04	22.409	2e−16	
Nduna pride	Intercept	5.81	0.85	6.86	7e−12	
distance to water	−1e−04	9e−05	−1.44	0.150	
shrub canopy cover	2e−04	3e−05	8.101	5e−16	
clay content	0.36	0.04	−8.851	2e−16	
soil depth	0.02	4e−03	−6.35	2e−10	
Hlamba mlonga pride	Intercept	−2.84	0.25	−11.28	2e−16	
distance to water	5e−04	4e−05	12.24	2e−16	
shrub canopy cover	−5e−05	2e−05	−3.02	0.003	
clay content	0.04	0.01	5.40	7e−08	
soil depth	0.02	2e−03	11.40	2e−16	
Note:

Presence probability ~ distance to water + shrub canopy cover + clay content + soil depth. Pr(>|z|) values <0.05 were considered significant.

Figure 4 Panel chart showing the relationship between the probability of lion presence and the logistic regression marginal effects of distance to water, shrub canopy volume, soil clay content and soil depth.

Home range selection appeared to be influenced by shrub cover, soil clay content and soil depth but responses to these variables were neither consistent among prides, nor aligned with the predicted effects of hierarchical dominance. The Banyini and Hlamba mlonga prides were positively associated with soil depth and clay content but negatively associated with shrub cover while the Nduna pride was positively associated with shrub cover and negatively associated with soil depth and clay content.

Influence of environmental factors on kill probability

For the Banyini pride, soil clay content (P < 0.01) and soil depth (P < 0.01) were negatively associated with kill probability while shrub canopy cover (P < 0.05) was positively associated with kill probability. Shrub cover (P < 0.01) and clay content (P < 0.05) positively influenced the Nduna and Hlamba mlonga prides respectively (Table 3). Distance to water was not significantly associated with kill probability for any of the prides (Table 3). Lion prides’ responses to the environmental variables measured were not consistent, for example kill probability was negatively and positively related to shrub canopy volume for the Banyini and Nduna prides, respectively (Fig. 5). Overall, the relationship between environmental variables and kill probability appeared to be weaker, as depicted by the large confidence intervals around the mean, compared to that with presence probability which had smaller confidence intervals (Figs. 4 and 5).

Table 3 Results of logistic regression analysis.

	Fixed effects	Estimate	Std. Error	z value	Pr(>|z|)	
Banyini pride	Intercept	4.53	1.19	3.66	>0.001	
distance to water	−6e−04	4e-04	−1.43	0.152	
shrub canopy cover	−2e−04	8e-05	−2.25	0.025	
clay content	0.18	0.04	−4.22	2e-05	
soil depth	0.03	0.01	2.76	0.006	
Nduna pride	Intercept	−4.00	1.56	−2.57	0.010	
distance to water	>0.01	>0.01	0.41	0.679	
shrub canopy cover	>0.01	>0.01	2.96	0.003	
clay content	0.05	0.03	1.59	0.249	
soil depth	0.01	0.01	1.15	0.112	
Hlamba mlonga pride	Intercept	−1.39	0.98	−1.41	0.158	
distance to water	1e−04	2e−04	0.69	0.490	
shrub canopy cover	−3e−05	8e−05	−0.34	0.738	
clay content	0.06	0.03	1.97	0.049	
soil depth	−4e−04	0.01	−0.06	0.950	
Note:

Kill probability ~ distance to water + shrub canopy cover + clay content + soil depth. Pr(>|z|) values <0.05 were considered significant.

Figure 5 Panel chart showing the relationship between kill probability and the logistic regression marginal effects of distance to water, shrub canopy volume, soil clay content and soil depth.

Discussion

Understanding where animals spend their time in an environment is a central tenet of ecology (Powers & McKee, 1994; Stephens & Krebs, 1986). The factors that underpin home range choice and hunting success among carnivores have been extensively explored (Brown & Kotler, 2004; Davies et al., 2016; Fryxell, 1991) yet new studies are warranted to broaden the body of knowledge. This study used GPS data to investigate the influence of environmental conditions and hierarchical dominance on the space-use and hunting success of three lion prides at MWR, Zimbabwe. Lion response was tested using soil depth, clay content, vegetation cover and distance to water as predictor variables. As primary building blocks of habitats, and determinants of forage and prey distributions, these variables are universal in every landscape and can thus be used to gauge expected lion behaviour where habitat-specific data are not available. Our findings showed that surface water was a key determinant of lion space-use and that pride size influenced home range selection among sympatric prides.

Factors influencing lion ranging and feeding ecology at MWR

Our study demonstrated that of the environmental variables measured, only distance from water reflected the predicted pride hierarchical response. Areas close to water are ordinarily associated with large aggregations of animals thereby facilitating higher prey encounter and kill rates (Redfern et al., 2003; Valeix et al., 2009). Our findings were consistent with the work of Owen-Smith (1996), Harrington et al. (1999), Valeix et al. (2010), de Boer et al. (2010), Cain, Owen-Smith & Macandza (2012), Duncan et al. (2012) and Kittle et al. (2022) who have also shown that lions preferentially select areas in which surface water is abundant. The ideal free distribution model (Fretwell & Lucas, 1969) has been widely used to demonstrate how resource availability and competition influence animal space-use where habitats vary. Our findings support our hypothesis that pride dominance determines home range selection among lions at MWR with the Banyini pride, which was the largest in size, occupying the central section of the reserve, which was well-endowed with water, while the Nduna (intermediate size) and Hlamba mlonga (smallest in size) prides selected the northern and southern sections with intermediate and low levels of surface water, respectively. Pride strength is a function of group size and the age-sex structure of a unit (Packer, Scheel & Pusey, 1990; Heinsohn, 1997). This is supported by earlier studies that document large prides routinely outcompeting smaller ones in inter-pride conflicts (Bertram, 1978; Packer, Scheel & Pusey, 1990) to occupy higher quality territories (Kauffman et al., 2007; Mosser et al., 2009). Where unrestricted, a pride is expected to select the most profitable portion of the landscape (Larson, 1980; Mitchell & Powell, 2004; Morris, 2003) and we speculate that had either the Nduna or the Hlamba mlonga pride had sole jurisdiction over the study area, they too would have selected the home range occupied by the Banyini pride.

Selection of shrub cover, soil clay content and soil depth attributes exhibited neither a consistent pattern nor followed hierarchical dominance in response to space-use by the three prides studied at MWR. As such, these variables were regarded lower order, and the associated responses inconsequential. We posit that once a home range has been selected based on the availability of surface water within the constraints of dominance hierarchy, a pride must make do with the configuration of lower order environmental attributes found in that habitat. At the resolution of our data, presence probability responses had smaller confidence intervals while kill probability responses had larger confidence intervals. This was likely because explanatory data for kills were not collected at the actual kill sites but were coarse averages derived at the scale of a vegetation map unit (see Clegg & O’Connor (2012)). Within a vegetation unit there are variations in shrub canopy cover, clay content and soil depth. Obtaining kill site specific measurements would have improved the resolution of these data but this was not feasible given the limited resources available for the study. Contrary to our hypothesis, proximity to surface water did not significantly influence kill probability. This outcome may be explained by prey species associating areas close to water with a higher predation risk and consequential elevated vigilance in these areas (Tuytens, 2019; Valeix, Chamaillé-Jammes & Fritz, 2007).

Where a generalized population-level response is required, it is common practice to pool observations from several groups belonging to the same population (Barker et al., 2023; Machlis, Dodd & Fentress, 1985). However, in this study it was observed that while the lions belonged to the same ecological system, the interaction of individual prides with their immediate environments differed and, in this case, pooling data would hide pride and habitat specific response patterns and produce flawed inferences. Knowledge of this effect is important because where it occurs, analyses of space-use that pools data across prides may yield spurious results. This is because pooling data assumes non-independence among intergroup observations and where such does not hold true the approach is likely to obscure functional differences that may exist between groups (Aebischer, Robertson & Kenward, 1993; Kuhar, 2006; Pollet et al., 2015). For example, an earlier analysis that was run using a pooled dataset produced a no relationship response between lion position data and distance from water, yet this was a key factor which lost significance when averaging directly opposing response patterns. Where lion ecology is studied and multiple prides are investigated, we recommend treating study prides as individual units. This is because the configuration of components and resources found in each pride’s home range vary and, likewise, each pride’s adaptations and interaction with its bounding habitat would be different. Where baseline information is not available, it is advisable to model ecological interactions from primary parameters (soil, water, vegetation) as secondary data (e.g., animal census statistics) may not be readily available or are costly to generate.

Management implications

This study shows that surface water availability is a key driver of lion space-use in semi-arid environments. The introduction of permanent water in previously waterless environments favours population growth of water-dependent animal species thereby improving conditions for lion success (de Boer et al., 2010; Kittle et al., 2022). For example, in Kruger National Park, South Africa the number of water points was increased across the park between 1939 and 1989 with a view to improving conditions for animals, but this triggered an increase in lion numbers precipitated by a proliferation of prey over a wider area of the landscape (Bryden, 1976; Grant et al., 2002; Harrington et al., 1999; Mills, Biggs & Whyte, 1995; Owen-Smith, 1996; Smuts, 1978). Access to prey is a key determinant of home range size among carnivores (Brown & Kotler, 2004; Loveridge et al., 2009; Mosser et al., 2009) and where prey is abundant (synonymous with habitats that are well-endowed with water), lion home range sizes are generally small and vice-versa (Joshi, Smith & Cuthbert, 1995; Macdonald & Carr, 1989; Mills & Knowlton, 1991).

Even though study prides did not show consistent responses to soil clay content, soil depth and shrub cover, the additive effect of these factors is important in defining habitat quality and, by extension, prey abundance and catchability (Hopcraft, Sinclair & Packer, 2005). Among the lower order variables, shrub cover is the one that can be easily manipulated by property managers either to open up or revegetate sections of veld depending on management’s goals. Earlier studies have shown that vegetation cover is an important factor in predator-prey interactions with both predators and prey using the cover of vegetation to their advantage, i.e., either to aid or evade predation (Van Orsdol, 1984; Spong, 2002; Hebblewhite et al., 2005; Hopcraft, Sinclair & Packer, 2005; Davidson et al., 2012; Minnie, Boshoff & Kerley, 2015). In this study, kills were recorded in both open and closed environments indicating that lions are adaptable in hunting strategy. This implies that in heterogenous environments management should maintain open and closed habitats as both are functionally important to the space-use and feeding ecology of lions.

The lion population at MWR follows a despotic distribution with the dominant pride inhabiting the area with the highest probability prey encounter and the most inferior pride occupying marginal environment where prey encounter rates were lowest. While marginal areas may be less profitable in terms of lion hunting, they are ecologically important to conservation managers as they serve as foraging grounds and refugia for species such as sable (Hippotragus niger) and Lichtenstein’s hartebeest (Alcelaphus lichtensteinii). Sable and hartebeest prefer taller grass which is usually found further from surface water due to reduced grazing pressure. In addition, the limited utilisation of these areas by water-dependent species results in reduced competition for food, low parasite loads and low predator presence (Cain, Owen-Smith & Macandza, 2012; Capon, Leslie & Clegg, 2013; Harrington et al., 1999). The existence and preservation of such habitats at MWR is considered to have contributed to the persistence of sable and Lichtenstein’s hartebeest’s populations on the property (Capon, Leslie & Clegg, 2013; Clegg et al., 2013).

Conclusions

This study has demonstrated that both social and environmental factors influence the ranging decisions and hunting success of lions at MWR. Most studies have focused on top-down processes of predator-prey interactions as the primary drivers and determinants of carnivore space-use choices. However, considering space-use and predation patterns in isolation from environmental and social factors does not provide a holistic understanding because ranging decisions are influenced by an interplay of biotic, abiotic and co-existence factors. Our findings confirm the hypothesis that the availability of surface water across the landscape influences space-use decisions of lions and shows that social dominance determines where in the landscape individual prides acquired their nutrition. We note that our sample size (n = 3 prides) was small and that collaring more prides would have improved the resolution of the data but resources were limiting. Repeating the experiment with a larger sample size and discrete habitat classes is the next step to producing habitat-specific responses and refined inferences for management.

We thank Colin Wenham for assistance with immobilizing target lions for collaring. We are indebted to Pandeni Chitimela, Julius Shimbani and the Malilangwe Scouts force for their assistance with field data collection.

Additional Information and Declarations

Competing Interests

Author Contributions

Animal Ethics

Data Availability

The authors declare that they have no competing interests.

Allan Tarugara analyzed the data, prepared figures and/or tables, authored or reviewed drafts of the article, and approved the final draft.

Bruce W. Clegg conceived and designed the experiments, performed the experiments, analyzed the data, authored or reviewed drafts of the article, and approved the final draft.

Sarah B. Clegg conceived and designed the experiments, performed the experiments, authored or reviewed drafts of the article, and approved the final draft.

The following information was supplied relating to ethical approvals (i.e., approving body and any reference numbers):

National Animal Research Ethics Committee of Zimbabwe provided full approval for this research (NAREC/008/23).

The following information was supplied regarding data availability:

The data for this study is available at Zenodo: Tarugara, A., Clegg, B., & Clegg, S. (2023). Factors influencing space-use and kill distribution of sympatric lion prides in a semi-arid savanna landscape. https://doi.org/10.5281/zenodo.10079892.

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
