# Peer review of "Factors influencing space-use and kill distribution of sympatric lion prides in a semi-arid savanna landscape"

_PeerJ, doi:10.7717/peerj.16749_

## Round 0.1 · original submission · Major Revisions

One reviewer states that the major shortcoming of this MS is that the sample size is too small (n = 3 prides; 50% of the prides in the reserve) to test the hypotheses and draw conclusions on the results. Given this, you need to carefully justify your tests and conclusions. Please also carefully revise other aspects of the paper to meet the criticisms of the reviewers.

·

Basic reporting

The authors have investigated if competition between lion groups/prides is affected by eco-geographical parameters and if that causes despotism in space use. The article is well written, however, I would specifically recommend not using the word 'subject' while addressing the lions in the study. I think that reduces such sentient animals to a level of automatons. Individuals or individual lions/groups are a better way to address them. I am not sure if 'game' is still a good word to use because it is very colonial. Using the word prey or prey species sounds more ecological.

The introduction would benefit from more context especially including the general overview of how superior prides become superior in the first place - it is not just the size of the pride but also the age of the females, and the number of resident adult males in the pride. Also a discussion of how competitively advanced prides have better reproductive success and thus higher recruitment, should be included. I think the idea of just considering pride size as an index of dominance is an oversimplification of a nuanced truth. There can be many pride members who are young and can actually be competitively outmatched by a smaller pride with more experienced adult lionesses. It is thus crucial to discuss this in the introduction, to set up the base for the study.

The table and figures are good, however, the dashed lines used for denoting MCPs of the prides are a bit confusing. Perhaps use starkly different colors? Are. you planning to hare the raw data?I understand this might be a stretch given the sensitive nature of the data - raw locations. Mentioning that might be useful

The introduction can benefit with well laid out predictions/hypothesis for the current study

Experimental design

The data collection is commendable however, I am looking for some clarifications regarding the experimental set up.
1. As mentioned earlier, considering only pride size is an oversimplification of a pride's competitive ability. What are the ages of the females in the pride? How many resident males in the coalitions for each pride? Since the prides are neighboring, did you observe any territorial strifes? If yes, who won those contests?
2. Did the same coalition of males range across multiple of these prides?
3. You mentioned that either a male was fitted with a collar or a female. Can you please include a breakdown of the number of collared animals in each pride? Or was it just 1 animal per pride? Males and females have distinct socio-biologies and hunting roles, and if the data is collected from a male in a pride vs a female in another, then they might not be comparable
4. Why did you not consider using Fixed Kernel methods of delineating core territories instead of 95% MCPs? MCPs can grossly overestimate the core areas of animals, especially where hunting/breeding happens, that are the most important when investigating competition
5. I am slightly confused about the random point generation approach. Will it not be more statistically and biologically meaningful to generate random points within their core/suitable habitats (equaling in number to the actual fixes) to delineate why an individual lion was found at a particular spot in its core habitat when that lion had a choice to be somewhere else in that habitat (which it uses)? Then the distance to water variable and other eco-geographical parameters make sense to compare between the actual fixes and the random points.
6. Are the fixes only from collars? Other group members of a pride were never located opportunistically or through monitoring?
7. Any comment on sub-group size of a pride? I presume all the individuals in a pride are not together at the same time. And studies show that sub-group size has a very strong effect on pride dynamics/cohesion/competitive ability. I wonder if you should include that a metric for competitive ability than pride size, sure worth considering.

Validity of the findings

I have added my thoughts about data analysis in the earlier section. Please refer to that.

Additional comments

I think this research has the promise to be a robust investigation of dominance in space use among animal groups, if the above comments are incorporated/addressed. I am really excited for the revisions.

·

Basic reporting

The article is well written (minor grammatical and syntax errors), structured and presented. Some additional background information can be provided to improve the context (see annotated MS). In addition, I have suggested some minor restructuring to improve the context of the introduction. The Authors provide appropriate references to existing literature. I have also made some minor suggestions on improving figures and tables. Finally, given the small samples size (n = 3 prides), it is difficult to test the ideal free hypothesis based on pride dominance. The authors attempt to justify this approach, but additional information on actual dominant-subordinate interaction will strengthen this approach (see annotated MS for relevant comments). In addition, the ideal free distribution hinges on variation in habitat quality, as aspect the Authors failed to address appropriately in the MS. As such, I have suggested an alternate approach in the attached annotated MS.

Experimental design

Given the small samples size (n = 3 prides), it is difficult to test the ideal free hypothesis based on pride dominance. The authors attempt to justify this approach, but additional information on actual dominant-subordinate interaction will strengthen this approach (see annotated MS for relevant comments). In addition, the ideal free distribution hinges on variation in habitat quality, as aspect the Authors failed to address appropriately in the MS. As such, I have suggested an alternate approach in the attached annotated MS.

Ethical clearance has been obtained.

There appears to be some more appropriate data available to test the hypotheses/predictions (see annotated MS). These may be more important in driving habitat selection and kill success for individual lion prides. In addition, some predictor variables are not ecologically relevant and collinearity between existing predictor variables was not assessed.

Validity of the findings

Given the conceptual and analytical approach, some of the conclusions cannot be justified (see annotated MS). This is compounded by the small sample size and lack of quantification of habitat quality (data on this is available; thus the stated assumptions may not be required). In addition, the results are not appropriately described and the authors glance over aspects. This results in some conclusions that are not supported by the results (e.g., kill locations).

Additional comments

Please refer to the annotated MS for additional comments.

---

## Round 0.2 · Minor Revisions

Please see the annotated manuscript and make the suggested changes.

·

Basic reporting

Fairly adequate revisions have been made, and the manuscript reads well. There are still words such as 'subjects' and 'game' that can be avoided (please see annotated pdf)

Experimental design

The number of prides from which data has been collected is small, especially because only 1 individual per pride was radio-collared. Thus, the entire ranging pattern of a pride hinges upon that one individual and based on if it was a male or a female, results can change quite a lot. I would have expected location data from other observations to be pooled in as well. If not, a discussion of this limitation is warranted. Pls see annotated pdf.

Validity of the findings

The study design can be bolstered by including other predictors beyond the ones used in this manuscript. Please see annotated pdf for comments

---

## Round 0.3 · accepted · Accept

Thank you for your careful attention to the reviewers' comments. I am now happy to accept the current manuscript for publication.